# The Literacy-Based Scale for Measuring Reflections on a University Social Responsibility Curriculum: Development and Validation

**DOI:** 10.3390/ijerph19084545

**Published:** 2022-04-09

**Authors:** Chia-Hui Hung, Cheng-Yi Huang, Yu-Ming Wang, Yi-Ching Li, Yung-Chuan Ho

**Affiliations:** 1Department of Occupational Therapy, Chung Shan Medical University, Taichung 40201, Taiwan; chhung@csmu.edu.tw; 2Department of Occupational Therapy, Chung Shan Medical University Hospital, Taichung 40201, Taiwan; 3Department of Nursing, Chung Shan Medical University, Taichung 40201, Taiwan; huangcy@csmu.edu.tw; 4Department of Nursing, Chung Shan Medical University Hospital, Taichung 40201, Taiwan; 5Department of Psychology, Chung Shan Medical University, Taichung 40201, Taiwan; 6Clinical Psychological Room, Chung Shan Medical University Hospital, Taichung 40201, Taiwan; 7Department of Pharmacology, School of Medicine, Chung Shan Medical University, Taichung 40201, Taiwan; annie@csmu.edu.tw; 8Department of Pharmacy, Chung Shan Medical University Hospital, Taichung 40201, Taiwan; 9Department of Medical Applied Chemistry, Chung Shan Medical University, Taichung 40201, Taiwan; ych065@csmu.edu.tw

**Keywords:** university social responsibility, literacy-based, service-learning, health promotion, scale development

## Abstract

University Social Responsibility (USR) enhances educational development and the impact of universities on society. As a stakeholder in USR, it is imperative to develop a comprehensive literacy scale that reflects the development of students’ citizenship in social engagement. Thus, this study aims to develop and validate the Health Promotion Literacy-based Scale for students in USR (HPLS-USR). A total of 200 students from USR with an average age of 19.27 participated in the study. The Exploratory Factor Analysis (EFA) was used to verify the scale’s construct validity. Twenty-two items were maintained in EFA with an internal consistency Cronbach’s α of 0.92. Construct validity was supported by EFA results, confirming that the four-factor structure of the 22-item scale (personal growth, responsibility of citizenship, social interaction, and intellectual growth) have reasonable correlations to each other, explaining 61.83% of the variance in the scale. The Kaiser–Meyer–Olkin index values of 0.908 and Bartlett’s Test of Sphericity (*p* = 0.001) verified the normal distribution of the EFA and the adequacy of the EFA sampling. These items achieved adequate factor loadings ranging between 0.44 and 0.82. This study demonstrated that the HPLS-USR has satisfactory construct validity and reliability in measuring students’ literacy abilities developed in USR participation.

## 1. Introduction

University Social Responsibility (USR) plays a critical role in enhancing educational development and facilitating the community’s quality of life owing to their education, research, and social actions [1,2]. The concept of USR is an extension of the spirit of corporate social responsibility (CSR), which is based on the pursuit of knowledge and truth, the cultivation of civic awareness, and the long-term development of society. The essence of a university is to assume social responsibility, and teachers, students, and administrators must respond through teaching, research, and public affairs participation so that the university can become a place of hope for future development [3,4,5]. Moreover, USR believes that university institutions should be involved in regional and global community services, sustainable economic development, ecological, social, environmental, and technical societal development. Responsible management of the academic learnings must implement the methods that affect labor, environmental development, teaching, research, service, and interaction with community and business environments.

An example of university social responsibility is the community regeneration of the beautiful but remote and declining traditional Paiwan indigenous tribe in eastern Taiwan. Through the integration of university and tribal resources, the cooperation and participation of university teachers, students, and residents to change the long-term disadvantage situation of the tribe in terms of the cultural, economic, and social parameters. University took care of the local community as their USR direction and combined traditional ecological and environmental wisdom into modern mountain and forest leisure activities. It creates experiential tours as a management method for economic development; the tribal residents have expectations for economic development and strive to preserve their culture and assist in the operation of tourism activities. Cooperation between the two parties established tribal tourism that benefits the tribe and the environment, and the tribe’s income has increased. This collaboration between the university and the tribe directly impacted the tribal tourism industry, resulting in indigenous people who had to work in other places to return to their hometowns and allowing foreign tourists to come into the tribe to understand the culture and environment. On the other side, students in the USR project could gain more professional learning experience by exploring local problems and needs, collecting data and brainstorming, and proposing solutions to solve the problem, thereby developing community-caring literacy of citizenship [6]. There was a similar example of university social responsibility in Malaysia. The university helped the neighborhood reduce social problems by providing facilities and services, and thereby enhanced their research energy and competitiveness [2].

A socially responsible university should design its vision and mission to benefit its environment and University Social Responsibility plays a critical role in enhancing educational development and facilitating the community’s quality of life owing to their education, research, and social actions [4,7]. Regarding the effectiveness of USR implementation, universities should construct student performance assessments to reflect the emerging needs of society in the context of globalization. However, the application of social responsibility in various industries is not the same, and stakeholders perceive the concept’s meaning of USR differently [8]. Therefore, different instruments are essential to measure the impact or change of a particular university’s social responsibility.

### 1.1. The Theoretical Basis of Literacy-Based Citizenship Developed in the USR Participation

Literacy is the internal quality that individuals develop and accumulates in their entire life. The manifestation of literacy as the external factor is indispensable from the knowledge, skills, and attitudes of a healthy individual to meet the needs of life situations [9]. United Nations Educational, Scientific and Cultural Organization, European Union, and The Organization for Economic Cooperation and Development have proposed the concept of core literacy, including lifelong learning, participation in society, and civic responsibility. Core literacy aims to develop high-quality citizens, promoting individual and social development; therefore, the concept is an important educational goal that universities must emphasize [9,10]. Dewey proposed that opportunities for experiential learning should be combined with teaching to enhance reflective action and allow individuals to continuously improve their knowledge, skills, and attitudes to develop new methods to deal with life’s complex problems [11,12]. Knowledge and skills could be acquired through teaching. However, attitudes are influenced by social interactions, including the people we encounter, historical experiences, and the cultural context in which we live, all of which are determinants of behavior [13,14,15].

This research is based on the social learning theory, which assumes that students acquire knowledge and skills, change attitudes, and develop civic literacy through participation in USR. Social learning refers to learning from interactions in a social context and is often used as the theoretical basis for attitude development [16]. Cognition plays a vital role in learning. People can learn by observing others and cognitively manipulating their social experiences to improve their intellectual skills. The internal insight process may influence attitudes and adaptive behavior [17]. Unlike the theory of rational behavior, which assumes that behavior is based on the control of an individual’s will, the theory of planned behavior (TPB) proposes a central principle for the internal process of insight, stating that their planned behavior determines an individual’s intention to act. In addition to attitude and subjective norms, TPB adds perceived behavioral control, which involves an individual’s ability to control their external environment, highlighting their experience and expected barriers that affect their perceptions and assessing the ease or difficulty of behaviors [18]. In this regard, universities must emphasize the importance of providing opportunities for social participation and experimental learning to develop their civic responsibility.

Literacy-based citizenship can be fostered through facilitating students’ social participation, nurturing the young workforce, and strengthening civic responsibility in volunteering services to corporations and the community [4,19]. Therefore, students are encouraged to participate in local community service-learning programs to promote health and well-being in rural and remote areas to reduce inequality in access to medical resources. Students will tackle challenges in a social context, improve critical thinking, problem-solving ability, resilience, self-efficacy, and emotional competence [4,20,21]. The mechanism of attitude change can be referred to as self-generated persuasion based on reflective practice through social participation in the courses, which promotes personal development [22]. Therefore, in this study, we identified four explicit attitudes factors, namely consciousness of self, relationship with others, intellectual abilities, and citizen’s responsibility attitude, as instructional objectives of the USR curriculum, which must be examined in understanding students’ development.

### 1.2. The USR Project and Curriculum in the Present Study

The USR Manifesto was launched in 2017 by the Higher Education Sprout Project in universities in Taiwan. This USR project aims to encourage universities to come up with humanities-oriented projects that consider local needs, provide local benefits for local connections, develop the local environment, solve problems facing the world, and improve the quality of life [7,20]. Five major themes were developed: Local Action and Connection, Industrial Collaboration, Environmental Sustainability, Food Safety, Long-Term Care, and other Social Practices. In 2017, Chung Shan Medical University launched the USR project of “Livable Shigang, Sustainable Health”. Shigang is a remote community in Taichung city in central Taiwan with a history of prosperity. Presently, the community faces population migration, aging, and unequal medical resources. In this project, we believe that through interdisciplinary teamwork, the medical service capacity provided by the medical university can provide preventive medical services for the community, health promotion, and achieve sustainable development goals (SDGs) [23]. With particular emphasis on SDGs in “no poverty”, “good health and well-being”, “quality education”, “responsible consumption and production”, “life below water”, and “life on land” to bring about positive changes to the community.

The USR project allows students to develop caring actions and connections in participation, learn to work altruistically with professionals in different disciplines, serve others, and achieve their learning goals by acquiring practical experience. In response to the community’s health issues and developing students, the College of Medicine integrates courses from different departments, including Nursing, Nutrition, Public Health, and Psychology, to develop academically participatory learning programs that educate students on how to contact the residents and participate in the community practice. These courses attempt to develop students’ literacies toward social responsibility in the community through physical, mental, and health promotion.

### 1.3. Developing the Assessment of Literacy in USR

An important issue of social responsibility research is measuring the effectiveness by examining stakeholders’ quality of life and the social impact created by the USR program [20,24]. In response, educators have explored theoretical and structural concepts or models that can be integrated with participatory curriculum to enhance students’ understanding of social responsibility, attitudes, awareness, and environmentally sound decision-making [25,26,27,28]. However, as stakeholders, students’ participation experience and literacy development in USR are equally important outcome indicators, but they have not been assessed in relevant research and must be addressed first [4].

Developing an integrated literacy scale is imperative to evaluate students’ civic learning by considering closeness, fairness, and integrity [29]. Currently, survey instruments, such as the self-report attitude scale, are widely used to evaluate students’ civic learning [4]. However, the USR curriculum is designed to augment in-depth comprehension of the subject matter and foster students’ awareness of complex social issues. Contrary to the community service that tends to be ill-defined, USR curriculums target specific course objectives, which link academic coursework with community service. USR curriculums also reflect the self role, active service, and professional skills to promote human and living creatures [1]. The experience acquired from USR would continue to retain, reproduce, and motivate students to social participation. Students could benefit from participating in USR civic learning, including improving their communication, learning motivation, engagement, and satisfaction. Therefore, college students’ civic literacy and social awareness development are critical parts of USR.

Due to the missions and educational goals of universities differences, a valid and reliable psychometric scale is crucial for a specific curriculum in a university. Thus, a questionnaire must undergo a robust development and validation process to ensure the credibility of the research findings. According to our knowledge, there is no relevance scale validated in Taiwan. This justifies developing and validating a new Health Promotion Literacy-based Scale in USR (HPLS-USR).

### 1.4. Overview of the Present Study

The present study aimed to develop a self-report measure, HPLS-USR, to measure students’ literacy in the USR curriculum. Two secondary goals are: (1) to design and development the HPLS-USR, and then to examine content validity, and (2) to assess psychometric properties to develop reliability and validity of the scale. The initial hypothesis was that the factor structure of the HPLS-USR would meet the behavioral parameters of explicit attitudes factors developed in USR. If the scale is developed, the next stage will be to measure attitudes and life reflections to maximize the utility of the scale in educational application. The development of the scale in this study can also fill the current lack of USR research on the effectiveness of student participation.

## 2. Materials and Methods

### 2.1. Research Design and Approach

This study employed a three-phase exploratory sequential mixed-method design to develop a prospective psychometric scale and construct the HPLS-USR. The first phase comprised item selection and development of an initial version of HPLS-USR. The second phase was dedicated to content validation, and the third phase validated the psychometric performance of HPLS-USR.

#### 2.1.1. Phase 1: Scale Design and Development

This phase began by drafting the HPLS-USR blueprint, based on a review of recent research on USR and service-learning evaluation, where significant literacies development was remarked in university students. This process helped identify several essential literacies development in the USR curriculum, assisting the authors in formulating, developing, and measuring a pool of 41 items. Of all these items, a response format was devised, based on a 5-point Likert scale (from 5: highly appropriate to 1: very inappropriate) to measure the agreement students developed in the USR programs [30,31,32].

#### 2.1.2. Phase 2: Content Validation

In this phase, a panel of three experts was asked to comment on the 41 items in the USR projects and determine the scale’s content validity. The experts were also asked to determine how they effectively measured relevant literacies, using three main criteria: (1) the relevance to perceive self-growth, community’s health need, literacies to civic responsibility, and practice. The other two criteria were (2) the clarity, adequacy, and length of wording, (3) the suitability and significance for the target population. Each panel member was asked to rate each item on a 4-point rating scale for content validity. The 4-point rating scale: 1 = not relevant, 2 = somewhat relevant, 3 = quite relevant, 4 = highly relevant was used to calculate the items of content validity index (CVI). The minimum 0.8 of CVI was considered valid [33,34]. Then, the panel member provided recommendations regarding any items that required changes or modifications. Panelists confirmed that 22 out of the original 41 items satisfied the three criteria and confirmed the face validity of the scale. The 22-item version of HPLS-USR proceeded to the psychometric testing.

#### 2.1.3. Phase 3: Psychometric Testing

In this phase, the reliability and validity of the scale were tested, then the Exploratory Factor Analysis (EFA) was applied for the factor analysis.

### 2.2. Study Population and Procedure

The samples comprised a respondent of undergraduate students who had experience participating in the USR programs. All participants obtained informed consent. Students are informed that participation in the study is voluntary and that there is no penalty for refusing to participate. Ethical approval was obtained from Chung Shan Medical University Hospital (Project no. CS2-21149).

The psychometric testing sample was estimated based on the 5:1 participants/data sets per item criterion [35]. A total of 207 respondents were eligible for the exam. Seven respondents were excluded due to insufficient data, and two hundred respondents participated in the complete data analyses.

### 2.3. Instruments

Respondents (*n* = 200) were asked to complete a socio-demographic form designed with HPLS-USR. Some respondents (*n* = 45) were asked to complete additional service-learning experiences scale and service-learning growth scale (SLES and SLGS) to determine the criteria-related validity [32].

The HPLS-USR is a self-reported questionnaire used to measure the experiences and reflections of students while taking USR curriculums. The scale consists of 22 items, each measuring specific reflections in USR courses. The items require participants to review and reflect on their learning experience and literacy development in USR courses. Items were scored using a 5-point Likert scale, with a minimum scale: 1 and a maximum scale: 5, represented never and always, respectively. The total scale score was calculated as the sum of all items in the Likert scale. Higher scores indicated a more pleasant experience.

The service-learning experiences scale and service-learning growth scale (SLES and SLGS) were developed by Chen, Pan and Shen in 2016 and consist of two scales with 35 items. Each item was rated using a five-degree self-assessment procedure to assess college students’ experience and growth in service-learning institutions. The scale was widely used in Taiwan [32]. In the absence of a scale or instrument to measure the effectiveness of USR participating, since the USR students’ participation is to some extent similar to service-learning, therefore, this study compared SLES and SLGS to establish criterion-related validity.

### 2.4. Data Analysis

The statistical analyses were performed using SPSS version 23 (IBM, Armonk, NY, USA). The psychometric analysis of the HPLS-USR was drawing on classical test theory (CTT), descriptive statistics were used to summarize the demographic data of the study participants and responses to each item. Internal consistency was estimated using Cronbach’s α coefficient. A reliability coefficient above 0.70 was considered acceptable for a new scale [33]. The item–total correlation was calculated using reliability analysis. A consistent scale, item–total correlation, and a corrected item–subscale correlation of 0.30 or higher was the criterion to select qualified items [36].

Construct validity was examined via the EFA to assess the constructive validity of the HPLS-USR. The Kaiser–Meyer–Olkin (KMO) test and Bartlett’s Test of Sphericity were used to determine sampling adequacy for the factor analysis. The criteria to determine the factor solution of factor extraction were (1) a factor with an eigenvalue of 1.00 or above, (2) an item with a factor loading of 0.40 or above, (3) no factor fewer than three items [37]. EFA is a statistical technique that examines the structure of latent variables (factors) under a set of observed variables (items). This approach helped to investigate if the internal structure of the scale of HPLS-USR fitted with the theoretical definition and conceptual framework [38,39] or could be used to test the theoretical model derived from EFA. Item analysis was applied to evaluate different item functioning.

In addition, criterion-related validity was estimated by examining Pearson correlations between the HPLS-USR and SLES and SLGS. The Pearson correlation coefficients were considered acceptable if they fell between medium (0.40–0.59) and strong (0.60–0.79). A *p* < 0.05 was considered the statistical significance level for all tests.

## 3. Results

### 3.1. Respondents Characteristics

Table 1 summarizes the sociodemographic data and educational characteristics of the 200 eligible respondents. More than half respondents (63.0%) were females, and the mean age of the total respondents was 19.27 (SD = 0.584) years. A total of 55.5% were first-grade undergraduates; 41.0% were second-grade undergraduates; 5.0% were third-grade undergraduates, and only 1 respondent (0.5%) was fourth-grade undergraduate. A total of 57.0% of respondents were in the school of Medicine and 43.0% in the school of Health Care and Management. All respondents had the experience of participating in service-learning, and most of them had already enrolled in the USR course (93.0%).

According to the analysis, the responses were considered a suitable representation of the population’s opinions for two reasons. First, most of the respondents had remarkable experiences in social participation. Second, the respondents reflected an excellent mix of grades in the university sample.

### 3.2. Descriptive Statistics and CTT Results for the HPLS-USR

Descriptive statistics for the 22 HPLS-USR items are displayed in Table 2. The overall mean scores were 87.52 (SD = 9.19). Responses to the items were overwhelmingly on the agree side of the item categories, with most items having 30% or more of the responses in the highest, “highly appropriate category”. The two highest response categories, which would represent the greatest level of each item, were recognized by respondents that their literacy meets the description of each item. Despite the low variability of responses across the five categories, the HPLS-USR demonstrated excellent internal reliability (Cronbach’s α was 0.92).

The results of the CTT analysis indicated that the alpha if deleted from each item, showed that overall reliability would not increase by removing any of the items. The item–total correlations, which reflect the correlation between an item and the total HPLS-USR score, ranged from 0.35 to 0.71, indicating good discrimination properties for each items.

### 3.3. Content Validity

Nine items were deleted in the first forty-one-item version scale because they did not meet the three main criteria for the primary selection. Then, an expert panel of three academic researchers in psychology, nursing, and occupational therapy evaluated the 22-item scale’s content. All items were evaluated for their CVI. Mean scores for each item ranged from 3.67 to 4.00. Based on the appraisal of the three experts [33], values that reached 1.00 were retained. The panel discussed and revised the items with a CVI of 0.8–0.99. The items with a CVI of less than 0.8 were removed from the scale. The I-CVI of each item of HPLS-USR all reached 1.00, and the S-CVI was 1.00.

### 3.4. Factor Structure of the HPLS-USR

Measurement adequacy was confirmed with the KMO = 0.908, and Bartlett’s test of sphericity confirmed that the factor structure was a good fit with the data (*χ*^2^(231) = 2308.19, *p* < 0.001). Based on the correlation matrix and communalities statistics, gradual elimination of the items with correlation less than 0.4. 

Table 3 shows the EFA results. The final EFA revealed a four-factor solution for the 22 items of the HPLS-USR that we retained. Given the use of a principal components analysis with varimax orthogonal rotation. We examined the structure and pattern coefficients to determine the most accurate factor structure.

We applied the four explicit attitudes factors mentioned earlier and our teaching objectives as indicators of literacy development to name the factors. Factor 1 (Personal Growth) included seven items (items 2, 3, 4, 5, 6, 7, and 8) that largely described the participant’s awareness of self-growth in USR—accounted for 39.80% of the variance. Five items in Factor 2 (Responsibility of Citizenship) (items 1, 15, 19, 21, and 22), that described participants’ concern for the community because they participated in USR, accounted for 11.50% of the variance. Factor 3 (Social Interaction) consisted of five items (items 9, 10, 13, 14 and 16) that described participants’ meaning about themselves because of engaging in interpersonal interactions, and accounted for 5.87% of the variance. Factor 4 (Intellectual Development) included five items (items 11, 12, 17, 18, and 20) that described the knowledge and skills acquired in the USR curriculum, and accounted for 4.66% of the variance. The four extracted factors comprised 61.83% of the total variance in HPLS-USR. Cronbach’s alpha coefficients for each factor was 0.90, 0.79, 0.81, and 0.79, respectively. The four factors are correlated to each other.

### 3.5. Correlations with the SLES and SLGS

The four factors of HPLS-USR significantly correlated with the SLES and SLGS in the current sample. The relationship between HPLS-USR and SLES and SLGS is summarized in Table 4. The factors, personal growth (*r* = 0.64, *p* < 0.001), responsibility of citizenship (*r* = 0.50, *p* < 0.001), social interaction (*r* = 0.76, p < 0.001), and intellectual development (*r* = 0.69, *p* < 0.001) in the HPLS-USR were positively correlated with SLES and SLGS. Finally, the HPLS-USR and SLES and SLGS offer a strong positive correlation (*r* = 0.76, *p* < 0.001).

## 4. Discussion

The study extends the existing research on USR, developing and validating the HPLS-USR to evaluate the formation of literacy in USR curriculums. Each step of the scale development was undertaken with scientific rigor. Items were initially generated based on the literature reviews and were tested through advanced psychometric techniques. The final 22-item scale measures respondents’ literacy in USR curriculums through four domains: personal growth, responsibility of citizenship, social interaction, intellectual development. These dimensions of the HPLS-USR align with the core goals of USR, that is, local connection and the cultivation of students’ literacy. 

Some of these dimensions are consistent with existing USR models, especially toward the stakeholder, the students, which supported by Latif’s attention to the dimensions of stakeholder responsibilities and development in USR [4]. The current findings suggested that personal factors such as personal growth and intellectual growth are consistent with attitude scales developed to measure the effectiveness and reflection of social participation [30,31,32]. Regarding civic literacy, environmental factors such as the responsibility of citizenship and social interaction are essential factors discussed in the social context of Taiwan [40,41]. In this study, the meaning ascribed to citizenship’s responsibility emerged as an expression of university responsibility [2], which was a school educational obligation to develop civic literacy. In addition, the emergence of social interaction was a critical context-specific dimension of USR that stressed the process, guided students’ engagement in interacting with the community, developing mutually beneficial relationships, and enhancing cooperation to improve the quality of life. Hence, these results provide strong evidence used in measuring literacy embedded within the social participation of USR.

### 4.1. Dimensions of the HPLS-USR

Dimensions of the HPLS-USR are part of factor analysis. The first factor, which is Personal Growth, based on the learning acquired, measures attitudes and awareness of individuals participating in USR health promotion field learning. A high score on this component indicates a stronger intention and greater likelihood of performing the target behavior in USR when the opportunity arises. USR curriculum can help students to broaden their knowledge, experience, and improve their ability to understand others and health problems. According to TPB, this aspect may come from self-awareness regarding the ability to control the external environment question “am I able to do it?”, thereby motivating (intention), improving ability (behavioral control), and enhancing behavioral achievement [42]. Previous reports suggested that if there were apperceived additional benefits of attendance in the class, students would engage in the learning process [42,43]. This factor is consistent with past studies that focus on the factor associated with social service-learning [31,32].

The second factor, Responsibility of Citizenship, consisted of items that described individuals who participated in field activities, and were sensitive to health problems and the community’s health. A high score on this factor indicates a high level of responsibility towards the community. Consistent with the expectations of college students from society, the responsibility of citizenship is an important attitude that must be developed in the social participation of USR curriculums [40,41]. Responsibility of citizenship comprises students’ active civic duties and skills needed to care for the community. This concept also focuses on the values of social justice, such as attitudes to poverty, social problems, and public policy, required by the service recipients [30]. However, the current study lacks such items in the scale, which must be supplemented in follow-up research.

The third factor, Social Interaction, consists of the items that highlight personal attitudes towards social caring, social responsibility, participation in public affairs, and respect for diversity. The items are based on two primary concepts: reciprocity and collaboration. A high score on this factor indicates that the learners perceive a relationship of equality and reciprocity with the residents of the community, in which both parties share responsibility, work together, and share results [44,45]. The USR curriculums provide students with meaningful community service participation and reflection. These also allow learners to have opportunities to connect academic learnings, civic roles, and to use their skills and knowledge to develop plans and strategies in addressing specific community problems and becoming actively contributing citizens. The association of social interaction and USR participation is consistent with much of the studies [30,31,32]. This factor is beneficial as it enhances social interaction skills, which is an important foundation for college students’ psychosocial development, self-identification, and a specific indicator of the students’ participation in USR.

The fourth factor, Intellectual Growth consists of items relevant to the individual’s cognitive learning, intellectual growth, and mental inspiration within the health promotion of the service-learning process. High scores on this factor recognized the knowledge and skills acquired in the USR curriculum. This factor is different from many outreached service-learning courses in USR curriculum as it facilitates the development of emotional attitudes, assists students to perceive cognitive learning, and enhances professional-related practical knowledge and skills. Based on the theory of social cognitive learning, the observation of good role models in social situations from teachers, peers, and residents will generate new behaviors [17]. Thus, the intellectual growth factor was stressed in USR curriculum especially in the Asian context [1,33].

### 4.2. Review of HPLS-USR

The 22-item HPLS-USR has passed the content validity test and EFA to form a reliable and valid scale that can reflect literacy development in USR participation. According to the aim of the study, this study has achieved the goals. However, given the current analysis of the measure, various recommendations are essential to consider. In the content validation stage, the expert panel finds that several of the items on the HPLS-USR appear to capture more than one literacy. For example, Item 14, “Through my field learning experience in the Shigang community, I have a more profound feeling about the relationship between people, community, and health”, contains an awareness of personal growth in addition to the recognition of social interaction. In a two-component question, one cannot determine the aspect of the question to which the respondent was responding. Further indicated by the strong relations between this item and both the approach and avoidance factors suggest that the item has components of each factor that may reduce the overall stability of the factor structure. This item, and others on the HPLS-USR, may benefit from editing to producing more substantial factors by separating the concepts from one another to improve the interpretability of the responses. Thus, it is recommended that the phrasing of certain items on the HPLS-USR be suitably modified to reflect specific coping strategies.

We obtained four factors from the EFA results that can be considered literacy developed in USR. However, according to the results of the criterion-related validity, the factor structure of HPLS-USR has a strong correlation with the service-learning scale [32], which also means that our scale should be adjusted in detail to make the scale more USR-specific. For example, USR’s participation involves cooperation with community residents. The concept of reciprocity can be subdivided from “social interaction”; the concept of social justice can be subdivided from “Responsibility of Citizenship” [30], so that the scale would provide student more specific feedback.

## 5. Implications of the Findings

As USR research continues to grow, it is vital to develop efficacy assessments to promote the reflection of literacies toward self-growth and citizenship participants within USR programs. HPLS-USR can help students reflect on themselves as it highlights the multidimensional nature of learning and participating in USR. It can support the research that focuses on attitude and behavior change, and then the literacy development in USR. The HPLS-USR consists of self-growth components, such as personal growth, intellectual growth, and components specific to the attitudes related to social influence, such as social interaction and the responsibility of citizenship.

The finding of this research provides further evidence for a more specific instrument to understand USR participants, especially students, and their development of civic literacy. Nevertheless, small samples of students are used for the learning effectiveness of participating in the USR programs. Research of a similar nature probably belongs to the learning experience and exploration of service-learning. The reflection on service-learning focuses on the growth of self-service experience. However, the expression of fairness, justice attitudes, and the degree of social influence of recipients are the spirits of USR. Thus, evaluation of the effectiveness of USR learning involves literacy development and reflection of the learning experience, which must conform to the nature of social needs and perspectives of different cultures.

The HPLS-USR was unique to the USR programs, as the items were developed and tested in the USR setting. As USR research is growing globally, the current research provides evidence for the validity of the HPLS-USR as a self-report measure to examine literacy developed in the USR programs. Data collection from this scale could help researchers and institutions to guide learning experience development and refine the USR curriculum. Although the scale was developed specifically for the university context, it can also be expanded in other sociocultural institutions for the application of a USR program.

## 6. Limitations and Suggestions for Future Studies

The study presents a rigorously validated literacy scale for the USR programs. We sampled enough participants from a university; however, most of the participants were from one university and had limited experience with USR. Future studies should extend the sample to other groups of interest to achieve more evidence for the validity of this newly developed scale. The small sample (*n* = 45) of the criterion-related respondents is used when generalizing the findings. Future research could expand upon the validation efforts of the HPLS-USR by revising the identified weak items and examining the factor structure with larger sample size and more heterogeneous characteristics. A confirmatory factor analysis would help to examine the dimensions and develop the model of the HPLS-USR. Further research is needed to examine and provide evidence for psychometric properties across the USR programs.

## 7. Conclusions

Developing citizenship through social engagement is an essential learning objective for students participating in USR. Previous research on USR has lacked the concern to measure the impact or change of stakeholders, especially with students. The present study has developed and tested the HPLS-USR, a reliable and valid tool with excellent psychometric properties for reflecting the literacy developed by the student in USR. The HPLS-USR contains 22 items in four factors, the four factors (personal growth, responsibility of citizenship, social interaction, intellectual growth) have reasonable correlations to each other. They can represent the literacy development in the USR. Thus, HPLS-USR may be used in future research projects worldwide because it correlates other service-learning and the capabilities that USR intends to develop in this regard. This study also proposes future research directions, including further studies to expand upon the validation efforts of the HPLS-USR by revising the identified weak items and examining the factor structure with a larger sample size and more heterogeneous characteristics. Furthermore, testing the confirmatory factor analysis would help examine the dimensions and develop the model of the HPLS-USR should be encouraged.

## Figures and Tables

**Table 1 ijerph-19-04545-t001:** Sociodemographic characteristics of the respondents.

Characteristics	Eligible Participants (N = 200)
	N (%) or Mean (SD)
**Age (year)**	19.27 (0.584)
**Gender**	
Men	74 (37.0)
Women	126 (63.0)
**Grade**	
1st	111 (55.5)
2nd	82 (41.0)
3rd	6 (5.0)
4th	1 (0.5)
**School**	
Medicine	114 (57.0)
Health care and management	86 (43.0)
**Participated in service-learning**	
Yes	200 (100)
No	0(0)
**Participated in USR (course)**	
1	186 (93.0)
>1	14 (7.0)

**Table 2 ijerph-19-04545-t002:** Descriptive statistics and CTT results for the HPLS-USR.

				Distribution of Item Responses (%)				
Items	Mean (*SD*)			Don’t Agree			Completely Agree	
		α If Deleted	Item–Total Correlation(r)	1	2	3	4	5
1. I will take the initiative to read books or information related to health promotion and social services.	3.49 (0.80)	0.92	0.54	1.0	7.5	42.0	40.5	9.0
2. I think taking this course that can help me develop my ability to explore health problems.	4.08 (0.65)	0.92	0.58	0	1.5	13.0	61.5	24.0
3. Participating in social service-learning for health promotion can broaden our knowledge.	4.37 (0.63)	0.92	0.60	0.5	0.5	3.5	53.0	42.5
4. Regular social service-learning can improve personal social experience.	4.37 (0.64)	0.92	0.56	0.5	0.5	4.0	52.0	43.0
5. Social service-learning for health promotion is beneficial to me.	4.28 (0.65)	0.92	0.71	0.5	0.5	6.5	55.5	37.0
6. It is right to serve the community and guide people to promote health.	4.38 (0.60)	0.92	0.55	0	0.5	4.5	51.5	43.5
7. Social service-learning is novel and interesting.	4.17 (0.70)	0.92	0.69	0.5	0.5	13.0	53.5	32.5
8. Participating in social service-learning in health promotion can help understand others better.	4.29 (0.61)	0.92	0.63	0	0.5	6.5	56.5	36.5
9. Promoting university social responsibility in Shigang enhances the connection between residents, the environment, and human values.	4.17 (0.62)	0.92	0.57	0	0.5	10.5	61.0	28.0
10. I will encourage others to adopt healthy habits and have regular physical examinations.	4.17 (0.67)	0.92	0.54	0.5	0.5	10.5	58.0	30.5
11. After completing the course, I can focus on community health issues from multiple perspectives.	3.96 (0.63)	0.92	0.71	0	1.0	18.5	64.0	16.5
12. I understand how to use scientific methods to assess field health issues.	4.09 (0.66)	0.92	0.42	0	0.5	16.0	58.0	25.5
13. To be a volunteer and promote health concepts make my life more valuable and meaningful.	4.16 (0.68)	0.92	0.71	0	1.0	13.5	54.5	31.0
14. Through my field learning experience in the Shigang community, I have a more profound feeling about the relationship between people, community, and health.	4.15 (0.62)	0.92	0.63	0	0	12.6	60.5	27.5
15. I began to learn about health promotion and events in the community where I live.	3.29 (0.84)	0.92	0.35	1.5	13.0	48.0	30.0	7.5
16. Through the field learning experience in the Shigang community, I have a better understanding of Taiwan’s rural communities.	4.06 (0.60)	0.92	0.57	0	0.5	14.0	65.0	25.0
17. When the residents of my community have difficulties, I will help them.	3.95 (0.63)	0.92	0.56	0	1.0	19.5	63.5	16.0
18. In the health promotion service, my learning motivation improved.	4.00 (0.65)	0.92	0.63	0	2.0	15.0	64.0	19.0
19. I began to feel that community health is the social responsibility of college students.	3.44 (0.82)	0.92	0.52	1.5	7.5	45.0	37.0	9.0
20. Participating in health promotion service activities allows me to find ways to contribute to society	3.90 (0.70)	0.92	0.65	0	2.5	22.0	58.0	17.0
21. I started to be sensitive to the health problems of the people.	3.42 (0.70)	0.92	0.46	0	8.5	45.0	43.0	3.0
22. I will take the initiative to care about people’s health needs.	3.36 (0.76)	0.92	0.52	0.5	8.5	52.5	31.5	7.0
Overall score of HPLS-USR	87.52 (9.19)							

**Table 3 ijerph-19-04545-t003:** Exploratory factor analysis: structure matrix coefficients for the HPLS-USR (N = 200).

Items	Factor Loadings			
	Factor 1	Factor 2	Factor 3	Factor 4
5. Social service-learning for health promotion is beneficial to me	**0.82**	0.20	0.13	0.18
3. Participating in social service-learning for health promotion can broaden our knowledge.	**0.80**	−0.04	0.26	0.09
6. It is right to serve the community and guide the people to promote health	**0.76**	−0.11	0.23	0.13
7. Social service-learning is novel and interesting.	**0.76**	0.27	0.13	0.17
8. Participating in social service-learning in health promotion can help understand others better.	**0.73**	0.01	0.02	0.27
4. Regular social service-learning can improve personal social experience.	**0.70**	0.15	0.17	0.18
2. I think this course can help me develop my ability to explore health problems.	**0.68**	0.27	0.26	−0.12
21.With this course, I began to be sensitive to health problems of people.	0.05	**0.75**	0.02	0.28
15. I began to learn about health promotion issues and events in the community where I live.	−0.06	**0.74**	0.23	0.01
22. I will take the initiative to care about people’s health needs.	0.13	**0.71**	−0.03	0.39
19. I began to feel that community health is the social responsibility of college students.	0.14	**0.60**	0.02	0.36
1. I will take the initiative to read books or information related to health promotion and social services.	0.29	**0.57**	0.31	0.02
16. Through the field learning experience in the Shigang community, I have a better understanding of Taiwan’s rural communities.	0.29	0.29	**0.67**	−0.04
10. I will encourage others to adopt healthy habits and have regular physical examinations.	0.35	0.05	**0.58**	0.19
13. Volunteering and promoting health concepts make my life more valuable and meaningful.	0.32	0.09	**0.55**	0.33
9. Promoting university social responsibility in Shigang enhances the connection between residents, the environment, and human values.	0.35	0.04	**0.47**	0.25
14. Through my field learning experience in the Shigang community, I have a more profound feeling about the relationship between people, community, and health.	0.37	0.14	**0.45**	0.28
12. I understand how to use scientific methods to assess field health issues.	0.26	0.11	0.14	**0.76**
17. When the residents of my community have difficulties, I will try to help them.	0.15	0.28	0.14	**0.74**
18. In the process of health promotion service, my learning motivation improved.	0.20	0.19	0.38	**0.68**
20. Participating in health promotion service activities allows me to find ways to contribute to society	0.33	0.37	0.17	**0.60**
11. After completing the course, I think I can focus on community health issues from multiple perspectives.	0.39	0.30	0.38	**0.44**
Factor correlations				
Factor 1	–			
Factor 2	0.36 *	–		
Factor 3	0.73 *	0.39 *	–	
Factor 4	0.57 *	0.62 *	0.63 *	–

Note. Factor 1 = personal growth; Factor 2 = responsibility of citizenship; Factor 3 = social interaction; Factor 4 = intellectual growth. Factor loading >0.4 are in boldface and are retained on the corresponding factor. * *p* < 0.001.

**Table 4 ijerph-19-04545-t004:** Correlations between the HPLS-USR and SLES and SLGS.

	Personal Growth	Responsibility of Citizenship	Social Interaction	Intellectual Growth	HPLS-USR
SLES and SLGS	0.64 *	0.50 *	0.76 *	0.69 *	0.76 *

Note. N = 45. HPLS-USR = the 22-item health promotion literacy-based scale in USR. SLES and SLGS = service-learning experiences scale and service-learning growth scale [32]. * *p* < 0.001.

## Data Availability

The datasets generated during the current study that support the findings of this study are available from the corresponding author upon reasonable request.

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
