# Peer review of "The Literacy-Based Scale for Measuring Reflections on a University Social Responsibility Curriculum: Development and Validation"

_ijerph, 2022, doi:10.3390/ijerph19084545_

Round 1

Reviewer 1 Report

Congratulations for responding to reviewers suggestions. Good luck,

  1. The Introduction: is complying to its scope.

In the Introduction section, has been improved.

  1. Literature background is standard but could be improved

Literature review has been improved

  1. Materials and Methods

This section has been improved

  1. Results

This section has been improved

5.Conclusions

This section has been improved

Author Response

Thank you for taking the time to review our manuscript and provide your comments.

Reviewer 2 Report

This is an intriguing paper and potentially useful to a portion of the IJERPH audience.  I would suggest though that the conceptualization of this aspect of social responsibility is a touch underdeveloped, and possibly assumes all readers have an understanding of the definitions, as well as the intention of offering such an initiative.  This paper could be useful more generally, beyond the specific case, with some effort to ground the conceptualization more generally in the literature and offer to readers a connection with their own work.  Also, the abstract is heavily analysis-driven and misses a potential opportunity to connect, again, with audiences beyond those with an interest already in this particular case. 

Author Response

Thank you for taking the time to review our manuscript and provide your comments.

According to your comments, we have:

(1) Adding concrete examples in the introduction section to the conceptualization of the USR (p2, lines 48-68).

(2) Revise the abstract and emphasize the introduction part (p1, lines 16-30).

This manuscript is a resubmission of an earlier submission. The following is a list of the peer review reports and author responses from that submission.

Round 1

Reviewer 1 Report

The research question is interesting and novel. There is an emerging focus on the topic of university social responsability thus the title is definitely attractive for the public.

About the Abstract: The abstract has a standard formulation.

  1. The Introduction: is complying to its scope.

In the Introduction section, the authors are revealing the purpose and the logic of the research, by briefly presenting the knowledge landscape on the subject and practical implications and also the theoretical and empirical research stages.

  1. Literature background is standard but could be improved

Literature review is detailed and complex as it debates all the concepts involved in the research.

In my opinion, from this literature review, and paper, misses the statement of hypothesis, research goal, objectives, importance or relevance of the research.

The authors provide only the description of the research, description of indices and indicators but no validation for any stated hypothesis.

  1. Materials and Methods

The presentation of the method is clear and very -well known by the audience, namely the Exploratory Factor Analysis (EFA).  The results and final are confusingly presented.

Tables are rather confusing and not structured well enough.

Please – rearrange Table 3.

IMPORTANT: General objective and main hypothesis must be formulated and discussed according to the research results.

  1. Results

Discussions section – is confusing and incomplete.

5.Conclusions

The text in the Conclusions section  is incomprehensible : ….

“This study was undertaken to develop a literacy-based scale for the implementation 409 of for use with participants in the USR programs. The current findings offer ais promising 410 valid argument about and indicates that the benefits of HPLS-USR, serve a is a psycho-411 metrically sound instrument and provide with a great deal of potential to build learning 412 efficacy research in the USR programs.”

Reviewer 2 Report

“Development and Validation of the Literacy-Based Scale for Measuring Reflections of University Social Responsibility Curriculum”. I believe the paper is not for publication.  explain below how the authors can improve this paper:

1. In the summary line 23 to 26 more emphasis is given to the statistical instrument used. The summary needs to be improved to help us better understand the objective of the work under study.

2. The introduction gives the impression that the USR was constructed from the university sphere, when its trajectory comes from the private sector with CSR. It is necessary to address this path and present the most relevant advances. Articles such as Garde Sanchez, R., Flórez-Parra, J. M., López-Pérez, M. V., & López-Hernández, A. M. (2020). Corporate governance and disclosure of information on corporate social responsibility: An analysis of the top 200 universities in the Shanghai ranking. Sustainability, 12(4), 1549, can help.

3. A section on the various theoretical frameworks of CSR and the theoretical framework on which the work is based would be missing. In addition, incorporating more elements without supporting it, such as the SDG approaches "The present study aimed to align with our educational goals and SDGs to construct a new Health Promotion-based".

4. In section 2 there are too many sub-sections to explain the different methodological elements. There is a need for unification. In addition, there are approaches without sufficient theoretical argumentation. For example, when talking about the Likert scale, there is no reference to it.

5. The results are too descriptive, it would be interesting to develop and/or point out some proposals related to the proposed objectives.

6. In the discussion, the analysis of literacy related to USR is taken very superficially and mixed with environmental indicators. Perhaps a better understanding of USR at the university level would help with the discussion, especially where theoretical frameworks would help in this regard.    

7. Although the article does point out some weaknesses with regard to the sample as it only takes students from one university. It would be useful to extend the sample to other groups of interest.

8. The conclusions are too short and need to be expanded.